**Data Availability Statement:** All DPC datasets have ethical or legal restrictions for public deposition due to inclusion of sensitive information

# Feasibility of management of hemodynamically stable patients with acute myocardial infarction following primary percutaneous coronary intervention in the general ward settings

**Kazuya Tateishi[1], Atsushi Nakagomi[2], Yuichi Saito[1], Hideki Kitahara[1], Masato Kanda[1], Yuki Shiko[3], Yohei Kawasaki[3], Hiroyo Kuwabara[4], Yoshio Kobayashi[1], Takahiro Inoue[4]\***

1 Department of Cardiovascular Medicine, Chiba University Graduate School of Medicine, Chiba, Japan,
2 Takemi Program in International Health, Harvard T.H. Chan School of Public Health, Boston, MA, United States of America, 3 Biostatistics Section, Clinical Research Center, Chiba University Hospital, Chiba, Japan,
4 Healthcare Management Research Center, Chiba University Hospital, Chiba, Japan

\* ifa8p8p@nifty.com

## Abstract

### Background

Although current guidelines recommend admission to the intensive/coronary care unit (ICU/CCU) for patients with ST-segment elevation myocardial infarction (MI), routine use of the CCU in uncomplicated patients with acute MI remains controversial. We aimed to evaluate the safety of management in the general ward (GW) of hemodynamically stable patients with acute MI after primary percutaneous coronary intervention (PCI).

### Methods

Using a large nationwide administrative database, a cohort of 19426 patients diagnosed with acute MI in 52 hospitals where a CCU was available were retrospectively analyzed. Patients with mechanical cardiac support and Killip classification 4, and those without primary PCI on admission were excluded. A total of 5736 patients were included and divided into the CCU (n = 3488) and GW (n = 2248) groups according to the type of hospitalization room after primary PCI. Propensity score matching was performed, and 1644 pairs were matched. The primary endpoint was in-hospital mortality at 30 days.

### Results

The CCU group had a higher rate of Killip classification 3 and ambulance use than the GW group. There was no significant difference in the incidence of in-hospital mortality within 30 days among the matched subjects. Multivariable Cox proportional hazard model analysis among unmatched patients supported the findings (hazard ratio 1.12, 95% confidence interval 0.66–1.91, p = 0.67).

from the human participants. Thus, all enquiries should be addressed to the data management committee via e-mail: byoin-kikaku@chiba-u.jp

**Funding:** The authors received no specific funding for this work.

**Competing interests:** The authors have declared that no competing interests exist.

## Conclusions

The use of the GW was not associated with higher in-hospital mortality in hemodynamically stable patients with acute MI after primary PCI. It may be feasible for the selected patients to be directly admitted to the GW after primary PCI.

## Introduction

Acute myocardial infarction (MI) remains a leading cause of death worldwide [1]. However, the prognosis of patients with acute MI has improved in recent decades and is mainly attributed to accessibility to early reperfusion therapy and established medical treatments [2, 3]. For example, between 1995 and 2015, 30-day mortality decreased in patients with ST-segment elevation myocardial infarction (STEMI) and non-ST-segment elevation myocardial infarction (NSTEMI) from 14% to 3% and from 11% to 3%, respectively, in France [2]. In-hospital mortality of acute MI in Japan has similarly decreased from 18.3% to 6.6% between 1980 and 2014 [4]. In particular, acute MI patients undergoing primary percutaneous coronary intervention (PCI) have better prognoses than their non-PCI counterparts, resulting in in-hospital mortality ranging from 2% to 6% [5, 6].

Management in an intensive care unit/coronary care unit (ICU/CCU) may also improve mortality in patients with acute MI through the monitoring of post-infarction complications, such as life-threatening ventricular arrhythmias or mechanical sequelae [7, 8]. However, the impact of CCU had been established before the reperfusion therapy era, although the incidence of complications has been greatly reduced by early reperfusion therapy, particularly primary PCI [9, 10]. It is unknown whether CCU is still beneficial to patients with acute MI undergoing primary PCI.

Recent European guidelines recommend using CCU for patients with STEMI [11], whereas American guidelines do not specify the role of CCU care in STEMI [12]. In Japanese guidelines, routine CCU use for patients with acute MI, including both STEMI and NSTEMI, is recommended [13]. Recent observational studies also showed mixed findings. Valley et al. reported that CCU rather than general ward (GW) admission for STEMI, but not NSTEMI, was associated with improved survival at 30 days [14]. Two studies indicated no impact of CCU use on clinical outcomes in patients with NSTEMI or low-risk patients with STEMI when compared to GW use [15, 16]. A possible explanation for this inconsistency is that the feasibility of patient care in the GW may vary by risk and severity of acute MI. Management in the GW may be acceptable for low-risk STEMI and NSTEMI. In the current era of early reperfusion therapy and established medical treatments, quite a few patients with acute MI could be managed safely in GW.

The impact of the CCU on in-hospital mortality should be evaluated using up-to-date data. In particular, more attention should be focused on patients with acute MI undergoing primary PCI because they are expected to have a low risk of in-hospital mortality. Therefore, we conducted this retrospective study to evaluate the safety of management of hemodynamically stable acute MI after primary PCI in the GW using large-scale data of patients admitted to hospitals from 2015 to 2019.

## Materials and methods

### Data source

The Diagnosis Procedure Combination (DPC) system is a case-mix classification system used in Japan to calculate reimbursements from insurers to acute care hospitals. This study used the

DPC database, which consists of administrative claims data regularly collected from voluntarily participating hospitals that operate under the DPC system. The DPC database includes summarized inpatient information, such as recorded diagnoses of the disease that resulted in hospitalization, other major diagnoses, Killip class on admission, comorbidities on admission, and discharge status. Diseases were identified through the International Classification of Disease, 10th revision (ICD-10), codes. The database also contains detailed information on the use of medical resources, diagnostic tests, surgical procedures, and prescribed medications. The Ethical Committee of Chiba University approved this study (unique identifier: 3309). As the data were anonymized, the requirement for informed consent was waived.

## Study population

We identified patients who fulfilled the following inclusion criteria: 1) age ≥18 years; 2) acute MI patients (ICD-10 codes: I21.0, I21.1, I21.2, I21.3, I21.4, and I21.9) admitted to a hospital between January 2015 and December 2019; and 3) patients treated at a hospital where a CCU was available (Fig 1). Patients were excluded from the present analysis based on the following criteria: 1) not receiving primary PCI on the day of admission; 2) receiving an intra-aortic balloon pump and/or extracorporeal membrane oxygenation on admission; 3) Killip class 4 on admission; and 4) missing outcome data (Fig 1). Briefly, patients with hemodynamically stable STEMI and NSTEMI were included. The final sample comprised 5736 patients (CCU: 3488 [61%]; GW: 2248 [39%]).

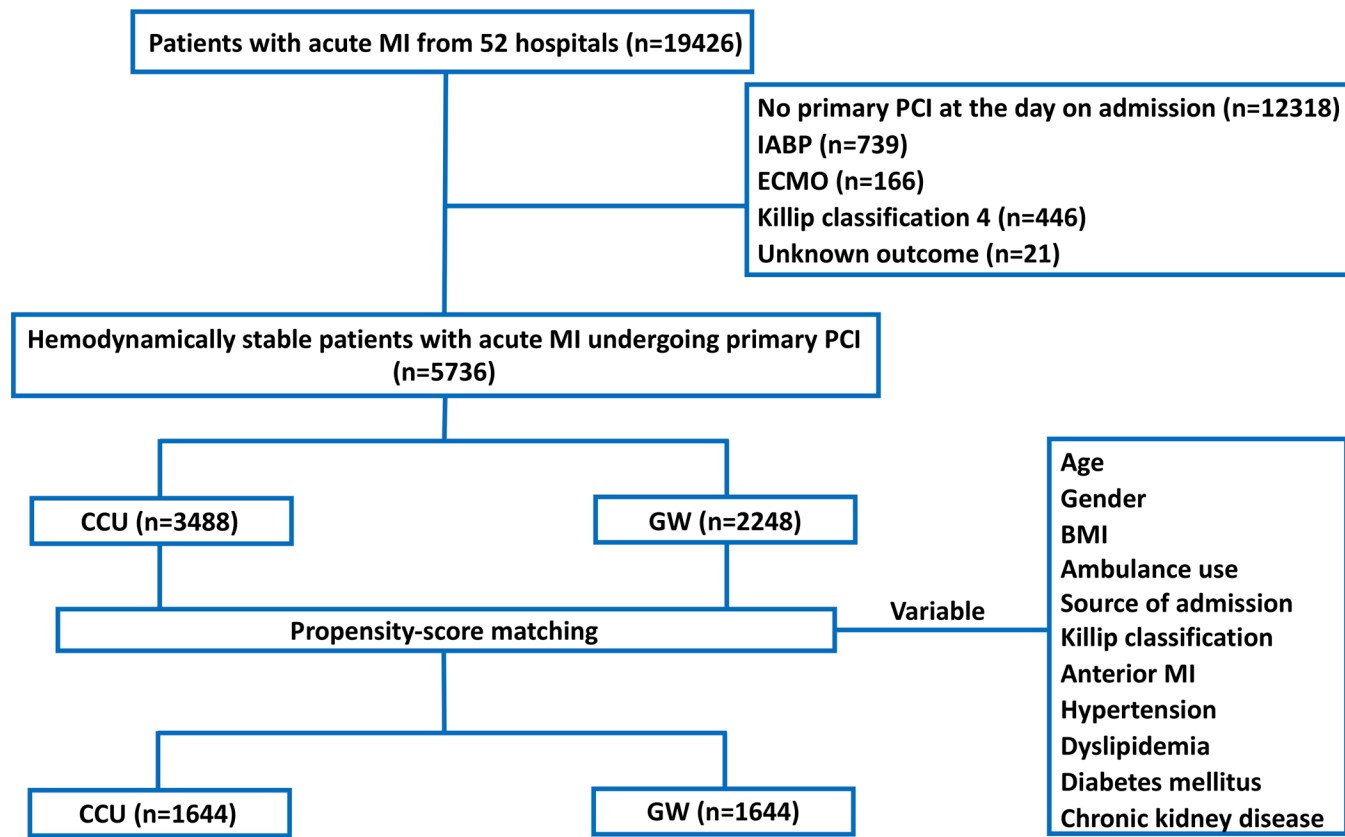

**Fig 1. Flowchart of the study population.** Emergency IABP and ECMO are defined as the procedures performed on the day of admission. BMI, body mass index; CCU, coronary care unit; ECMO, extracorporeal membrane oxygenation; GW, general ward; IABP, intra-aortic balloon pump; MI, myocardial infarction; PCI, percutaneous coronary intervention.

## Outcomes

The primary outcome was in-hospital mortality within 30 days, measured from the time of hospital admission. We also calculated the total duration of hospitalization (days from admission to discharge) and total hospitalization cost based on reimbursement of treatment costs from the DPC system.

## Variables

The exposure variable was CCU use on hospital admission. Admission to the CCU was defined as the presence of an ICU/CCU revenue center code in the administrative DPC data. We defined GW admission as the lack of any ICU/CCU revenue center code. Patients were divided into CCU and GW groups according to the type of hospitalization room on admission after primary PCI. We included the following variables as confounding factors: age, sex, ambulance use, source of admission (home, another hospital, and nursing home), body mass index, Killip classification on admission, the presence of anterior MI, and comorbidities (hypertension, dyslipidemia, diabetes mellitus, and chronic kidney disease). The use of an intra-aortic balloon pump and extracorporeal membrane oxygenation beyond the day of admission, ventilator, intravenous vasopressor agent (catecholamine, etc.) and intravenous vasodilators (nitrates, carperitide, etc.) were selected as surrogates of disease severity. We also evaluated medications, including antiplatelet agents, renin-angiotensin system blockers, β-blockers, and statins, during hospitalization to examine compliance with medical treatments.

## Statistical analysis

All statistical analyses were performed using the SAS statistical software package version 9.4 (SAS Institute, Cary, USA). Continuous variables were expressed as mean ± standard deviation when normally distributed, and as median and interquartile range when non-normally distributed. Categorical data were presented as absolute numbers and percentages. Continuous variables were compared using Student's t-test or Mann-Whitney U-test, as appropriate. Categorical variables were compared using Pearson's chi-square test or Fisher's exact test. Propensity scores were calculated for each participant using multivariate logistic regression based on the variables shown in Fig 1. We conducted a 1:1 propensity score matching (PSM) analysis without replacement (greedy matching algorithm), with a caliper width equal to 0.2 of standard deviation of the logit of the propensity score. To examine the balance of covariate distribution between the CCU and GW groups, we calculated the standardized difference (SD). The Kaplan-Meier method and log-rank test were used to analyze the 30-day survival in PS-matched groups. As a sensitivity analysis, in-hospital mortality through hospitalization was used as an outcome. Furthermore, univariate and multivariate Cox proportional-hazards regression analyses for 30-day mortality and in-hospital mortality during hospitalization were performed using the unmatched patients. We adjusted for patient age, sex, body mass index, prevalence of anterior MI, use of ambulance, source of admission, and Killip classification on admission, and determined the hazard ratio (HR) and 95% confidence intervals (CIs) for each variable. P value <0.05 was considered statistically significant.

## Results

The clinical characteristics of the study population are summarized in Table 1. The CCU group had a higher rate of ambulance use and Killip class 3, whereas transfer from another hospital and unclassified Killip subgroups were more frequently found in the GW group. After PSM, the two matched groups (1644 patients in each arm) showed no significant differences in baseline clinical characteristics, including ambulance use and Killip classification (Table 1).

**Table 1. Patient characteristics (before and after propensity score-matching).**

| Variable | Before Propensity score matching | | | After Propensity score matching | | |
|---|---|---|---|---|---|---|
| | CCU (n = 3488) | GW (n = 2248) | SD | CCU (n = 1644) | GW (n = 1644) | SD |
| Age (years) | 68.6±12.8 | 69.4±12.8 | −0.063 | 68.7±12.7 | 68.9±12.9 | −0.013 |
| Male | 2700 (77%) | 1747 (78%) | −0.007 | 1305 (79%) | 1298 (79%) | 0.011 |
| BMI (kg/m$^2$) | 24.0±3.8 | 24.1±3.8 | −0.008 | 24.0±3.8 | 24.0±3.8 | −0.019 |
| <18.5 | 167/3313 (5%) | 101/1983 (5%) | −0.002 | 84 (5%) | 87 (5%) | −0.008 |
| 18.5–25.0 | 1956/3313 (59%) | 1172/1983 (59%) | −0.001 | 969 (59%) | 968 (59%) | 0.001 |
| >25.0 | 1190/3313 (36%) | 710/1983 (36%) | 0.003 | 591 (36%) | 589 (36%) | 0.003 |
| Ambulance use | 2492 (71%) | 1500 (67%) | 0.102 | 1136 (69%) | 1139 (69%) | -0.004 |
| Source of admission | | | | | | |
| Home | 3328 (95%) | 2103 (94%) | 0.082 | 1543 (94%) | 1538 (94%) | 0.013 |
| Another hospital | 118 (3%) | 126 (6%) | -0.107 | 84 (5%) | 95 (6%) | -0.030 |
| Nursing home | 42 (1%) | 19 (1%) | 0.035 | 17 (1%) | 11 (1%) | 0.039 |
| Killip Classification | | | | | | |
| 1 | 2179 (63%) | 1294 (58%) | 0.100 | 1093 (66%) | 1072 (65%) | 0.027 |
| 2 | 965 (28%) | 513 (23%) | 0.112 | 426 (26%) | 442 (27%) | −0.022 |
| 3 | 261 (7%) | 85 (4%) | 0.161 | 68 (4%) | 73 (4%) | −0.015 |
| Unclassified | 83 (2%) | 356 (16%) | −0.481 | 57 (3%) | 57 (3%) | 0.000 |
| Anterior MI | 1486/3103 (48%) | 925/1994 (46%) | 0.030 | 769 (47%) | 749 (46%) | 0.024 |
| Hypertension | 1250 (36%) | 813 (36%) | −0.007 | 618 (38%) | 611 (37%) | 0.009 |
| Diabetes mellitus | 1134 (33%) | 716 (32%) | 0.014 | 513 (31%) | 517 (31%) | -0.005 |
| Dyslipidemia | 2389 (68%) | 1500 (67%) | 0.038 | 1124 (68%) | 1118 (68%) | 0.008 |
| Chronic kidney disease | 190 (5%) | 126 (6%) | −0.007 | 65 (4%) | 81 (5%) | −0.048 |

Data are shown as mean ± standard deviation, or number (%). BMI, body mass index; CCU, coronary care unit; GW, general ward; MI, myocardial infarction; SD, standardized difference.

Medications, such as antiplatelet agents, β-blockers, angiotensin-converting enzyme inhibitors, and statins, were well prescribed in both CCU and GW groups before and after PSM (Table 2). There was no significant difference in crude 30-day in-hospital mortality before and after PSM between CCU and GW groups (before: 1.7% vs. 1.8%, p = 0.93; after: 1.8% vs. 1.1%, p = 0.08).

Kaplan-Meier curve analysis after PSM showed no significant difference in the incidence of 30-day in-hospital mortality (Fig 2). Sensitivity analysis applying multivariate Cox proportional hazard model analysis by using the unmatched patients supported the findings (HR 1.12, 95% CI 0.66–1.91, p = 0.67), showing that older age, admission from nursing home, and Killip classification were significantly associated with an increased risk of 30-day in-hospital mortality (S1 Table). Similar patterns were observed even when in-hospital mortality during entire hospitalization period was used as outcome (PSM analysis: log rank p = 0.17; Cox proportional-hazards regression analysis: HR 1.02, 95% CI 0.63–1.69, p = 0.91).

In addition, we examined the length of stay and total hospitalization costs. The total duration of hospitalization was shorter (11 [9–15] vs. 12 [9–16] days, p<0.001), and total hospitalization cost was lower in the GW group than in the CCU group after PSM (1780000 [1480000–2150000] vs. 1830000 [1560000–2240000] JPY, p<0.001)

## Discussion

In the present study, we focused on low-risk patients with acute MI: patients with primary PCI and without mechanical circulatory support on arrival. As expected, the 30-day mortality was

**Table 2. Post treatment variables and outcomes (before and after propensity score-matched cohort).**

| Variable | Before Propensity score Matching | | | After Propensity score Matching | | |
|---|---|---|---|---|---|---|
| | CCU (n = 3488) | GW (n = 2248) | P value | CCU (n = 1644) | GW (n = 1644) | P value |
| Medication | | | | | | |
| Antiplatelet agent | 3423 (98%) | 2235 (99%) | <0.0001 | 1612 (98%) | 1638 (99.6%) | <0.0001 |
| Statins | 3189 (91%) | 2093 (93%) | 0.021 | 1491 (91%) | 1538 (94%) | 0.002 |
| ACEI/ARB | 2695 (77%) | 1872 (83%) | <0.0001 | 1276 (78%) | 1422 (87%) | <0.0001 |
| β-blocker | 1582 (45%) | 1105 (49%) | 0.005 | 725 (44%) | 826 (50%) | 0.0004 |
| Intravenous vasopressors | 95 (3%) | 92 (4%) | 0.005 | 42 (3%) | 67 (4%) | 0.015 |
| Intravenous vasodilators | 322 (9%) | 383 (17%) | <0.0001 | 166 (10%) | 253 (15%) | <0.0001 |
| Ventilator | 52 (1.5%) | 21 (0.9%) | 0.061 | 20 (1%) | 18 (1%) | 0.744 |
| IABP | 124 (4%) | 67 (3%) | 0.233 | 53 (3%) | 31 (2%) | 0.015 |
| ECMO | 8 (0.2%) | 6 (0.3%) | 0.780 | 5 (0.1%) | 3 (0.2%) | 0.726 |
| Hospitalization day, days | 12 [9–16] | 11 [8–15] | <0.0001 | 12 [9–16] | 11 [9–15] | <0.0001 |
| Hospitalization cost, 10,000 JPY | 184 [156–224] | 176 [144–217] | <0.0001 | 183 [156–224] | 178 [148–215] | <0.0001 |
| 30-day mortality | 61 (1.7%) | 40 (1.8%) | 0.932 | 30 (1.8%) | 18 (1.1%) | 0.079 |
| In-hospital mortality | 68 (1.9%) | 48 (2.1%) | 0.627 | 34 (2.1%) | 22 (1.3%) | 0.105 |

Data are shown as mean ± standard deviation, median (interquartile range), or number (%). ACEI, angiotensin-converting-enzyme inhibitors; ARB, angiotensin receptor blockers; CCU, coronary care unit; GW, general ward; IABP, intra-aortic balloon pump; PCPS, percutaneous cardiopulmonary support.

lower in this population, with no significant difference between CCU and GW groups (1.7% vs. 1.8% in the CCU and GW groups). To our knowledge, this is the first report investigating the safety and feasibility of management in the GW of hemodynamically stable patients with acute MI following primary PCI.

The 30-day mortality of acute MI in this study (1.7% vs. 1.8% in the CCU and GW groups) was low, as expected. Two previous studies reported no impact of CCU use on clinical outcomes in patients with NSTEMI with low in-hospital mortality (1.3% vs. 1.2% in the CCU and GW groups) and STEMI patients with low APACHE III scores and a low likelihood of complications [15, 16]. On the other hand, Valley et al. showed that ICU admission was associated with improved survival at 30 days among patients with STEMI with high mortality (14.3% vs. 8.3% in the ICU and GW groups) [14]. This may explain the survival benefit of the ICU in high-risk cohorts, whereas low-risk patients (i.e., patients with low in-hospital mortality) may be managed appropriately in the GW. Our study also showed no association between ICU admission and in-hospital mortality among the matched patients (very low-risk patients: 30 days mortality, 1.8% vs. 1.1% in the CCU and GW groups) and the unmatched patients (the entire population) in the sensitivity analysis, supporting the feasibility of management in the GW for low-risk patients with acute MI.

CCU use is potentially beneficial for patients with cardiogenic shock (Killip 4) or requiring mechanical circulatory support, who were excluded from this study. On the other hand, hemodynamically stable patients undergoing PCI could be managed safely in the GW, probably because the rate of life-threatening ventricular arrhythmias and mechanical complications in acute MI has been largely reduced by early reperfusion therapy, particularly by primary PCI [9, 10], with a reported incidence of 0.27% in STEMI and 0.06% in NSTEMI [17]. Indeed, the reported in-hospital mortality rate of patients with acute MI following primary PCI is quite low: 2.2%-6.1% among European countries and 2.7%/0.7% in patients with STEMI/NSTE-ACS in Japan [5, 6]. Outcomes of acute MI in the current era have entailed such low rates of in-hospital mortality to suggest that routine use of the CCU rather than the GW may

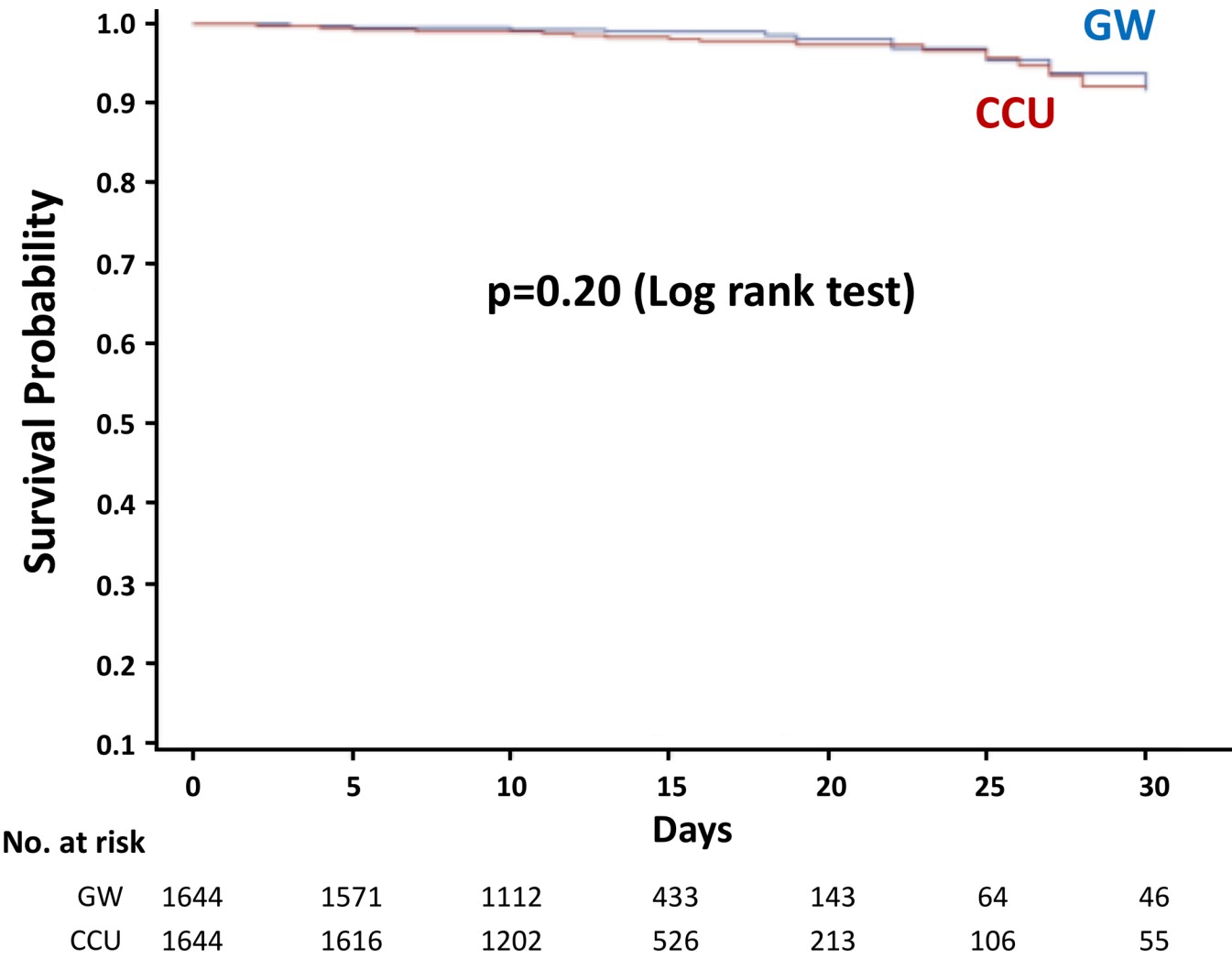

**Fig 2. Kaplan–Meier curves for the probability of survival after propensity score matching.** CCU, coronary care unit; GW, general ward.

not provide significant survival benefits, especially in hemodynamically stable patients. In addition to early reperfusion therapy, optical medical therapy (i.e., antiplatelet agents, β-blockers, angiotensin-converting enzyme inhibitors, and statins), as was done in our study, could be vital for safe management of the selected patients in the GW [18]. Further studies are needed to evaluate the criteria for admission to the CCU and essential factors, such as optical medical therapy, in the safe management of the selected low-risk patients with acute MI in the GW.

Primary PCI has been widely performed in patients with acute MI (France: 76%, Japan: 85%) [2, 19, 20]. In the era of primary PCI, care of selected low-risk populations with acute MI in the GW may be beneficial in terms of not only in-hospital mortality, but also the length of hospital stay and total hospitalization costs. We found that the total hospitalization costs of direct admission to the GW were lower than those of admission to the CCU. CCU care is a significant component of total costs worldwide [21]. Previous studies have reported that mean hospitalization costs were 2.5 times higher among patients admitted to the ICU than those managed in the GW [21, 22]. In addition, it is conceivable that management of hemodynamically stable patients with acute MI in GW offers opportunities for other patients who require intensive care in CCU. Considering the limited capacity of ICU beds, this would be beneficial

to improve health resource allocation. Although this study did not focus on cost effectiveness, our findings warrant further investigation about the potential benefit of GW management of acute MI patients to save cost and medical resources.

## Limitations

There were several limitations to the present study. First, this was a retrospective study using the DPC administrative database, which does not provide detailed clinical information including past medical history, laboratory findings, reperfusion strategies, and ST-segment elevation on electrocardiogram. Thus, residual confounding based on these factors cannot be ruled out. In the present study, acute MI includes STEMI and NSTEMI, both of which could not be distinguished. Second, we excluded patients with acute MI who underwent primary PCI on the day following admission or later. This may have impacted patient selection. However, given that immediate or early invasive strategies are recommended in patients with STEMI and high-risk NSTEMI [11], patients who did not undergo primary PCI within 24 hours may belong to different subsets of acute MI, such as supply/demand (type 2) MI. Third, patient care in the GW was left to the discretion of physicians in daily clinical practice; therefore, it could not be defined systematically. Fourth, the very low-risk patients were selected among all patients by PSM, and treated with acute MI. However, sensitivity analysis applying the multivariable Cox proportional hazard model also supported the outcome in all subjects. Lastly, PCI procedures done in Japanese routine clinical practice are different from those in Western countries in some aspects, including the predominant use of intracoronary imaging in Japan [23–25].

## Conclusions

The present large-scale data set demonstrated that using the GW instead of the CCU was not associated with an increased risk of in-hospital mortality in hemodynamically stable patients with acute MI after primary PCI. The total duration of hospitalization was shorter, and total hospitalization cost was lower in the GW group. Patient care in the GW directly after primary PCI may be feasible and safe in selected populations of acute MI.

## Supporting information

**S1 Table. Multivariable analysis for factors associated with 30 days in-hospital mortality.** BMI, body mass index; CCU, coronary care unit; CI, confidence interval; HR, hazard ratio; MI, myocardial infarction. Age, gender, Ambulance use, Killip classification, Anterior MI and using CCU on admission were entered into a multivariable model.
(DOCX)

**S1 Checklist. The RECORD statement–checklist of items, extended from the STROBE statement, that should be reported in observational studies using routinely collected health data.**
(DOCX)

## Acknowledgments

We thank all of the hospital staff who assisted in data collection.

## Author Contributions

**Conceptualization:** Kazuya Tateishi, Atsushi Nakagomi, Yuichi Saito, Hideki Kitahara, Masato Kanda, Yoshio Kobayashi, Takahiro Inoue.

**Data curation:** Kazuya Tateishi, Masato Kanda, Yuki Shiko, Yohei Kawasaki, Hiroyo Kuwabara, Takahiro Inoue.

**Formal analysis:** Yuki Shiko, Yohei Kawasaki, Hiroyo Kuwabara.

**Supervision:** Yoshio Kobayashi, Takahiro Inoue.

**Validation:** Takahiro Inoue.

**Writing – original draft:** Kazuya Tateishi.

**Writing – review & editing:** Atsushi Nakagomi, Yuichi Saito, Hideki Kitahara, Yoshio Kobayashi, Takahiro Inoue.

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
