## [Decision Letter · Decision Letter 0]

7 Sep 2020

PONE-D-20-25203

Feasibility of management of hemodynamically stable patients with acute myocardial infarction following primary percutaneous coronary intervention in the general ward settings

PLOS ONE

Dear Dr. Inoue,

Thank you for submitting your manuscript to PLOS ONE. After careful consideration, we feel that it has merit but does not fully meet PLOS ONE’s publication criteria as it currently stands. Therefore, we invite you to submit a revised version of the manuscript that addresses the points raised during the review process.

We look forward to receiving your revised manuscript.

Kind regards,

Corstiaan den Uil

Academic Editor

PLOS ONE

Journal Requirements:

2. As part of your revision, please complete and submit a copy of the RECORD checklist, a document that aims to improve reporting and reproducibility of observational studies  that use routinely-collected data for purposes of post-publication data analysis and reproducibility (http://record-statement.org). Please include your completed checklist as a Supporting Information file. Note that if your paper is accepted for publication, this checklist will be published as part of your article.

Reviewers' comments:

Reviewer's Responses to Questions

**Comments to the Author**

1. Is the manuscript technically sound, and do the data support the conclusions?

Reviewer #1: Yes

2. Has the statistical analysis been performed appropriately and rigorously? 

Reviewer #1: Yes

3. Have the authors made all data underlying the findings in their manuscript fully available?

Reviewer #1: Yes

4. Is the manuscript presented in an intelligible fashion and written in standard English?

Reviewer #1: Yes

5. Review Comments to the Author

Reviewer #1: In this study, Tateishi and colleagues used an Japanese nationwide admin dataset to examine pts with N/STEMI admitted to 52 hospitals to CCUs or wards in a propensity matched retrospective cohort. After adjustment, they reported no differences in 30-day mortality. The study is important insofar as it adds more information to the growing knowledge that CCU is not required for uncomplicated Mis. A more selective admission practice also has the potential to reduce health care costs. I offer the following comments

1. What ist he accuracy of diagnostic codes form STEMI in your dataset?

2. Where there differences between STEMI vs NSTEMI pts?

3. In the STEMI cohort, were there difference by delays to revascularization, primary PCI vs lytics, or success of reperfusion?

4. Interhospital transfers may represent a sicker cohort of patients. How were these patients handed in the analysis?

5. Do the authors have any information on patients requiring intravenous vasoactive agents or ventilation?

6. What were the potential cost savings if this was applied more broadly?

6. PLOS authors have the option to publish the peer review history of their article (what does this mean?). If published, this will include your full peer review and any attached files.

Reviewer #1: No

---

## [Author Response · Author response to Decision Letter 0]

22 Sep 2020

Responses to the comments of Reviewer #1

Thank you for the careful and comprehensive review of our manuscript. In response to the Reviewer’s comments and recommendations, we have answered the questions in a point-by-point fashion and revised our manuscript as follows. Changes we made are written in red in the manuscript.

1. What is the accuracy of diagnostic codes form STEMI in your dataset?

2. Where there differences between STEMI vs NSTEMI pts?

> Because of the lack of information in the DPC database, it is impossible to identify patients with STEMI/NSTEMI. We acknowledge that this is the one of major limitations of our study. Thus, we have added the following sentence in the Limitation section.

Page 12, Line 22-23

“In the present study, acute MI includes STEMI and NSTEMI, both of which could not be distinguished.”

3. In the STEMI cohort, were there difference by delays to revascularization, primary PCI vs lytics, or success of reperfusion?

> In-hospital mortality in patients with STEMI depends on initial strategies (i.e. timely primary PCI, late primary PCI, or a pharmaco-invasive strategy) as shown in the FAST-MI program [Eur Heart J. 2020;41:858-66]. We believe that this is of clinical interest. However, because of the lack information in the DPC database, STEMI patients could not be identified in the present study. Thus, we have amended the following sentence in the Limitation section. 

Page 12, Line 18-21

“First, this was a retrospective study using the DPC administrative database, which does not provide detailed clinical information including past medical history, laboratory findings, reperfusion strategies, and ST-segment elevation on electrocardiogram.” 

4. Interhospital transfers may represent a sicker cohort of patients. How were these patients handed in the analysis?

> According to the reviewer’s suggestion, we have added the data into Table 1 and 2. When simply dividing patients into two groups (admission from home vs. transfer from another hospital), there was no significant difference in 30-day mortality (1.7% vs. 2.5%, p=0.325). However, we found that patients admitted from nursing home had significantly higher mortality. Therefore, we divided patients into three groups (admission from home vs. transfer from another hospital vs. admission from nursing home), resulting in significant difference in 30-day mortality (1.6% vs. 2.5% vs. 16.4%, p<0.001). Accordingly, we have included the source of admission as a confounding factor into Cox proportional hazard model and re-conducted PSM analysis. Nevertheless, the basic results did not change. We have amended the Methods and Results sections, and Table 1, 2, S1, Figure 1 and 2. 

Page 2, Line17-18

“hazard ratio: 1.12, 95% confidence interval 0.66 - 1.91, p=0.67”

Page 6, Line 7-13

“We included the following variables as confounding factors: age, sex, ambulance use, source of admission (home, another hospital, and nursing home), body mass index, Killip classification on admission, the presence of anterior MI, and comorbidities (hypertension, dyslipidemia, diabetes mellitus, and chronic kidney disease).”

Page 7, Line 10-11

“source of admission”

Page 7, Line 16

“transfer from another hospital”

Page 8, Line 7

“after: 1.8% vs. 1.1%, p=0.08”

Page 9, Line 8-9

“HR 1.12, 95% CI 0.66-1.91, p=0.67”

Page 9, Line 9

“admission from nursing home”

Page 9, Line 12

“PSM analysis: log rank p=0.17; Cox proportional-hazards regression analysis: HR 1.02, 95% CI 0.63-1.69, p=0.91”

Page 10, Line 7

“11 [9-15] vs. 12 [9-16] days, p<0.001”

Page 10, Line 8-9

“1780000 [1480000-2150000] vs. 1830000 [1560000-2240000] JPY, p<0.001”

Page 11, Line 5-6

“1.8% vs. 1.1%”

5. Do the authors have any information on patients requiring intravenous vasoactive agents or ventilation?

We appreciate the reviewer’s meaningful comment. We have added the information of intravenous vasoactive agent (vasopressor and vasodilator agents) in Table 2. Also, we have added the data regarding ventilation in Table 2. There was no significant difference in the use of ventilator between the GW and CCU groups. On the other hand, the GW group had higher incidence of use of intravenous vasopressor and vasodilator agents than CCU group before and after PSM. We have amended the Methods section accordingly.

Page 6, Line10-13

“The use of an intra-aortic balloon pump and extracorporeal membrane oxygenation beyond the day of admission and ventilator, intravenous vasopressor agents (catecholamine, etc.) and intravenous vasodilators (nitrates, carperitide, etc.) were selected as surrogates of disease severity.”

6. What were the potential cost savings if this was applied more broadly?

In this present study, 5736 out of 19426 patients (29.5%) with acute MI were hemodynamically stable, and 3488 of 5736 patients (60.8%) were admitted to CCU rather than GW. Total hospitalization cost was 80000JPY higher in the CCU group than in the GW group in this study. Considering the number of patients diagnosed as acute MI each year in Japan (n=82000), if all hemodynamically stable patients with acute MI defined in the present study were initially managed in GW, 1176560000JPY (approximately 11 million US dollars) might be saved in Japan each year, although this is a very rough estimate and may be beyond the scope of the present study. We have added the following sentences in the Discussion section. 

Page12, Line9-12

“In addition, it is conceivable that management of hemodynamically stable patients with acute MI in GW offers opportunities for other patients who require intensive care in CCU. Considering the limited capacity of ICU beds, this would be beneficial to improve health resource allocation.”

---

## [Editor Report · Decision Letter 1]

25 Sep 2020

Feasibility of management of hemodynamically stable patients with acute myocardial infarction following primary percutaneous coronary intervention in the general ward settings

PONE-D-20-25203R1

Dear Dr. Inoue,

We’re pleased to inform you that your manuscript has been judged scientifically suitable for publication and will be formally accepted for publication once it meets all outstanding technical requirements.

Kind regards,

Corstiaan den Uil

Academic Editor

PLOS ONE
---

## [Editor Report · Acceptance letter]

30 Sep 2020

PONE-D-20-25203R1 

Feasibility of management of hemodynamically stable patients with acute myocardial infarction following primary percutaneous coronary intervention in the general ward settings 

Dear Dr. Inoue:

I'm pleased to inform you that your manuscript has been deemed suitable for publication in PLOS ONE. Congratulations! Your manuscript is now with our production department. 

Kind regards, 

on behalf of

Dr. Corstiaan den Uil 

Academic Editor

PLOS ONE